# Pressure-induced reversal of Peierls-like distortions elicits the polyamorphic transition in GeTe and GeSe

Tomoki Fujita [1,10], Yuhan Chen [2,10], Yoshio Kono [3], Seiya Takahashi[4], Hidetaka Kasai [4], Davide Campi[5], Marco Bernasconi [5], Koji Ohara[6], Hirokatsu Yumoto [7,8], Takahisa Koyama [7,8], Hiroshi Yamazaki[7,8], Yasunori Senba[7,8], Haruhiko Ohashi[7,8], Ichiro Inoue[8], Yujiro Hayashi[8], Makina Yabashi [8], Eiji Nishibori [4], Riccardo Mazzarello [2] ✉ & Shuai Wei [1,9] ✉

While polymorphism is prevalent in crystalline solids, polyamorphism draws increasing interest in various types of amorphous solids. Recent studies suggested that supercooling of liquid phase-change materials (PCMs) induces Peierls-like distortions in their local structures, underlying their liquid-liquid transitions before vitrification. However, the mechanism of how the vitrified phases undergo a possible polyamorphic transition remains elusive. Here, using high-energy synchrotron X-rays, we can access the precise pair distribution functions under high pressure and provide clear evidence that pressure can reverse the Peierls-like distortions, eliciting a polyamorphic transition in GeTe and GeSe. Combined with simulations based on machine-learned-neural-network potential, our structural analysis reveals a high-pressure state characterized by diminished Peierls-like distortion, greater coherence length, reduced compressibility, and a narrowing bandgap. Our finding underscores the crucial role of Peierls-like distortions in amorphous octahedral systems including PCMs. These distortions can be controlled through pressure and composition, offering potentials for designing properties in PCM-based devices.

If a liquid is cooled rapidly enough and crystallization is avoided, it can solidify into an amorphous rigid state (i.e., glass). While the phenomenology of liquids and glasses has been generally established, there are systems (e.g. water and silicon) that exhibit anomalous behaviors[1,2]. They have two distinct liquid states and two corresponding amorphous solids, characterized by remarkably different structures and physical properties (e.g., density, entropy, electric, and rheological properties) while maintaining identical chemical composition. The transition between the two liquid phases is known as liquid–liquid transition (LLT). The solid-state counterpart of the LLT is instead called

[1]Department of Chemistry, Aarhus University, 8000 Aarhus C, Denmark. [2]Department of Physics, Sapienza University of Rome, Rome 00185, Italy. [3]Geodynamics Research Center, Ehime University, Matsuyama 790-8577, Japan. [4]Department of Physics, Faculty of Pure and Applied Sciences and Tsukuba Research Center for Energy Materials Science (TREMS), University of Tsukuba, Ibaraki 305-8571, Japan. [5]Department of Materials Science, University of Milano-Bicocca, I-20125 Milano, Italy. [6]Faculty of Materials for Energy, Shimane University, Matsue, Shimane 690-8504, Japan. [7]Japan Synchrotron Radiation Research Institute, 1-1-1 Kouto, Sayo-cho, Sayo-gun, Hyogo 679-5198, Japan. [8]RIKEN SPring-8 Center, 1-1-1 Kouto, Sayo-cho, Sayo-gun, Hyogo 679-5148, Japan. [9]iMAT Centre for Integrated Materials Research, Aarhus University, Aarhus, Denmark. [10]These authors contributed equally: Tomoki Fujita, Yuhan Chen. ✉e-mail: riccardo.mazzarello@uniroma1.it; shuai.wei@chem.au.dk

polyamorphic transition[2]. The first and most well-known example of a compound exhibiting both transitions is water[3–5], but there are several other materials that show similar transitions including silica, silicon, germanium, molecular liquids triphenyl phosphite, and even some metallic glass-formers[6–10].

There is growing interest in the LLT and polyamorphism in the class of functional phase-change materials (PCMs) (e.g. Ge–Sb–Te, GeTe). PCMs can be rapidly and reversibly switched between the amorphous and crystalline states by electrical or optical pulses via Joule or laser heating. Taking advantage of the strong optical and electrical contrast between the different states, information can be encoded in each state for nonvolatile data storage and neuromorphic computing applications. A recent study, using the femtosecond diffraction technique with X-ray free electron lasers, reported direct evidence of an LLT in the supercooled liquid of two PCMs, namely $Ag_4In_3Sb_{67}Te_{26}$ and $Ge_{15}Sb_{85}$[11]. It was suggested that the LLT is relevant to the switching speed, property contrast, and amorphous stability[12], as the LLT is not only a high- to low-density transition[13] accompanied by a fragile-strong transition with a drastic change in viscosity (as in water), but also a metal-to-semiconductor transition[14]. Recent computational work identified the temperature of the metal-to-semiconductor transition in GeTe and $Ge_2Sb_2Te_5$ in the deeply supercooled liquid below the melting temperature $T_m$[15]. In contrast to other anomalous tetrahedral liquids (e.g., water and Si), in PCMs, the LLT is governed by temperature-induced Peierls-like distortions in the local structure. The latter refers to distortions of the $p$-bonded octahedral coordination, where alternating short and long bonds form on the opposite sides of a central atom, lowering the total energy and opening a pseudo-bandgap[16].

Given the existence of two liquid phases, their vitrification may lead to two distinct glasses inheriting the structure and properties of their respective liquid. Earlier studies reported drastically different structures of amorphous PCM samples of $Ge_2Sb_2Te_5$ prepared through different processing routes[17,18]. Several studies observed pressure- or temperature-induced "polyamorphic" transitions in $Ge_2Sb_2Te_5$, $Ge_1Sb_2Te_4$, and $K_2Sb_8Se_{13}$[19–22]. As the resistivity drop associated with the transition can be of a few orders of magnitude, thus comparable to the drop brought about by crystallization, the polyamorphic transition has also attracted attention for improving the device stability or designing multi-level memories with high storage density[21,23]. Nonetheless, it remains unclear if the polyamorphic transition mechanism in PCMs can be understood within the same theoretical framework as their LLTs. In particular, is it possible to attribute the polyamorphic transition to alterations in Peierls-like distortions? If this is the case, which factors can be adjusted to exert control over the Peierls-like distortion and consequently modify the polyamorphic transition behavior?

In this work, we address these open questions by focusing on the pressure as the thermodynamic variable to alter the amorphous state of the two isoelectronic $p$-bonded compounds GeTe and GeSe. Both compounds exhibit (defective) octahedral-like atomic arrangements with varying levels of distortions[24]. While GeTe is a typical PCM with fast crystallization kinetics and peculiar bonding mechanisms in the crystalline state[25], GeSe displays a more covalent character with slower crystallization kinetics[24]. We conducted in-situ high-pressure pair distribution function measurement using a high-flux pink beam at the photon energy of X-ray of 100.1298 keV at the BL05XU beamline in SPring-8. The technique enabled us to precisely determine structure factor $S(Q)$ of amorphous GeSe and GeTe at the wide range of $Q$ up to 27 Å⁻¹ under in-situ high-pressure conditions, which opens a new way to investigate pressure-induced structural change. We also performed molecular dynamics simulations employing a machine-learned neural network potential (NNMD) to study the behavior of large models with atomic-level details. We report the pressure-induced polyamorphic transition in GeSe and GeTe and provide evidence that the suppression of Peierls-like distortions elicits the transition. The structural

characteristics of the low- and high-pressure states are identified through the analysis of total structure factors and pair distribution functions. The simulations support the experimental findings and suggest a semiconductor-to-metal transition in conjunction with the closing of the pseudo-bandgap. Our findings show that pressure can effectively reverse the Peierls-like distortions that form during the metal-to-semiconductor transition upon supercooling at ambient pressure. This suggests that multiple pathways, involving various pressure–temperature combinations, can be used to manipulate the Peierls-like distortions and tailor the material properties accordingly.

## Results

### Reversing the Peierls-like distortion and polyamorphic transition

We performed an in-situ high-pressure synchrotron X-ray diffraction experiment for the as-deposited amorphous GeSe and GeTe at the BL05XU beamline in SPring-8 (see the "Methods" section). Conventional high-pressure scattering experiments can only provide a rather limited momentum transfer $Q$-range due to the limited scattering angle of high-pressure cells and relatively low energy of ~30-40 keV. Here we take advantage of a high energy 100.1298 keV photon beam combined with Paris-Edinburgh press, which enabled us to collect the diffraction data with a large momentum transfer $Q$-range up to 27 Å⁻¹ (Ref. 26). This extensive $Q$-range allows us to extract high-resolution pair distribution functions (PDF) in real space, which are relevant for the comprehensive examination of Peierls-like distortions within the local atomic environment.

Figure 1a and b show the measured diffraction intensity $I(Q)$ of amorphous GeSe and GeTe, respectively, at a pressure ranging from ambient pressure to ~10 GPa. The broad scattering peaks of the amorphous states are observed at 2.11 Å⁻¹ and 3.59 Å⁻¹ for GeSe and at 1.99 and 3.38 Å⁻¹ for GeTe at ambient pressure. With increasing pressure, the sharp Bragg peaks of GeTe appear at 3.4 GPa, indicating partial crystallization, while GeSe remains fully amorphous up to 10.0 GPa. The crystallized phase of GeTe is identified as the rock-salt type cubic phase (space group $Fm\bar{3}m$) by the Rietveld refinement, as shown in the inset of Fig. 1b. The crystallization pressure is close to the pressure of the rhombohedral-to-cubic transition of crystalline GeTe (3.5 GPa)[27], while no crystallization of GeSe is observed up to 10.0 GPa. This difference between GeTe and GeSe can be ascribed to their distinct bonding characteristics and local structural configurations, which will be further discussed later (see the "Discussion" section). After the pressure release, the cubic phase of GeTe is transformed into the stable rhombohedral phase (see Supplementary Fig. 2b).

GeSe and GeTe show a pre-peak of $I(Q)$ around $Q = 0.95$ Å⁻¹ and $Q = 0.8$ Å⁻¹, respectively (Fig. 2a and b). As demonstrated in earlier studies, a pre-peak for the $p$-bonded (defective) octahedral systems is a footprint of the Peierls-like distortion[11,28]. The peak corresponds to the quasi-periodic structure stemming from the formation of alternating long and short bonds, resulting in the medium-range structural order with roughly twice the periodic distance of the original octahedral arrangement. The pre-peaks shift toward a higher $Q$ and become smaller under compression. Figure 2c shows the pressure dependence of the integrated intensity of the pre-peaks. After normalizing the intensity $I_{ppk}$ to the result at ambient pressure $P_{amb}$, the data points can be fitted with a stretched exponential decay $\frac{I_{ppk}(P)}{I_{ppk}(P_{amb})} = \exp\left\{-\left(\frac{P}{\alpha}\right)^\gamma\right\}$ for both of GeSe and GeTe, suggesting the suppression of the Peierls-like distortion by pressure. The fitting parameters are $\alpha = 2.6 \pm 0.3$ GPa and $\gamma = 0.89 \pm 0.06$ for GeSe, and $\alpha = 0.4 \pm 0.2$ GPa and $\gamma = 0.36 \pm 0.09$ for GeTe, demonstrating the significantly faster decay of the pre-peak in GeTe. The pre-peak of GeTe vanishes at 3.4 GPa, while the pre-peak of GeSe disappears at 9.1 GPa.

To investigate the structural evolution associated with the Peierls-like distortion, we have extracted the total structure factor $S(Q)$ from

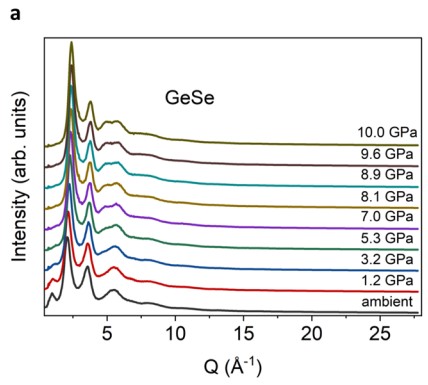

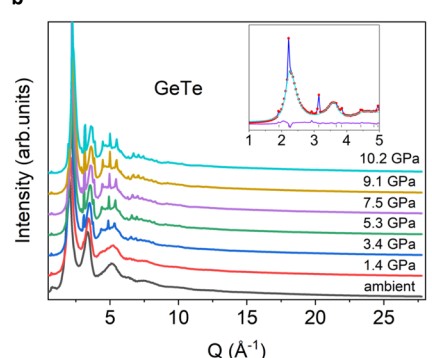

**Fig. 1 | The in-situ X-ray diffraction intensity $I(Q)$ under compression from the ambient pressure to ~10 GPa. a** GeSe and **b** GeTe The GeSe remains fully amorphous up to 10.0 GPa, while GeTe is partially crystalized at 3.4 GPa. The inset of **b** is the result of the Rietveld refinement of GeTe at 7.5 GPa with the experimental data (red dots), the calculated data of the cubic GeTe (blue line), the background (light blue line), and the residuals between experimental and calculated data (purple line).

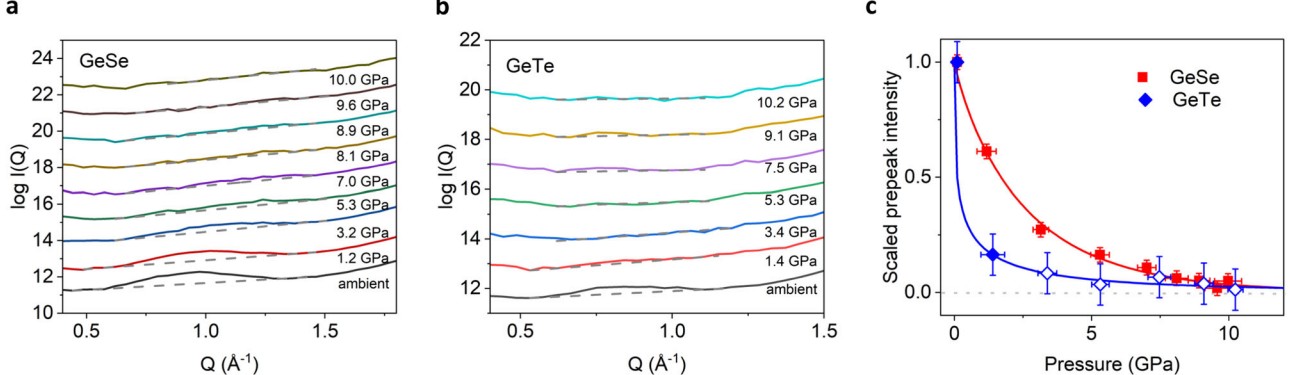

**Fig. 2 | The pre-peak of the diffraction intensity $I(Q)$ in the logarithmic scale. a** GeSe and **b** GeTe. The $Q$-range from 0.4 to 1.8 Å⁻¹ is shown for a clear view of the pre-peaks. The gray dashed lines are the baselines calculated by a linear interpolation[62]. **c** The pressure dependence of the integrated intensity of the pre-peak of GeSe (square) and GeTe (rhombus). The open symbols of GeTe above 3.4 GPa indicate that the system contains a partially crystallized state at those pressures. The pre-peak intensity of each sample is normalized to the intensity at the ambient condition, respectively. Error bars were estimated for the selection of data points for linear interpolation, which is significantly larger than the error caused by the fluctuation of $I(Q)$ itself. Red and blue lines are the fitting by a stretched exponential function for the guide of eyes.

$I(Q)$ and Fourier transformed it to the reduced pair distribution function (PDF) G($r$). The details of the data processing are explained in the Supplementary Notes (see also "Methods" section, Fig. 3a, b, and Supplementary Figs. 1–3). The PDF analysis allows us to obtain directly the real-space structural information. In amorphous PCM systems, it has been revealed that the formation of alternating long and short bonds by the Peierls-like distortion elongates the bond length of some atoms in the first coordination shell close to the second coordination shell, thereby making it difficult to resolve these two coordination shells[11]. However, the ratio of the second to the first peak positions of $G(r)$, $R = r_2/r_1$, has been shown to be a useful structural parameter to characterize the extent of the Peierls-like distortion in liquids or amorphous solids[11,29].

Figure 3c shows the pressure dependence of $R$. The $R$ of GeSe decreases from 1.59 at ambient pressure to 1.43 at 10.0 GPa. This is the opposite behavior to the change of $R$ during the temperature-induced LLT in the supercooled liquid state of $Ge_{15}Sb_{85}$ and $Ag_4In_3Sb_{67}Te_{26}$, where $R$ increases from 1.36 to 1.50 accompanied by the emergence of the pre-peak[11]. The pressure dependence of $R$ of GeSe can be fitted by an error function, while the fit is difficult for GeTe due to the limited number of data points at low pressure (Fig. 3c). The pressure derivative of $R$, $-dR/dP$, is shown in Fig. 3d, indicating that the decrease rate of $R$ keeps increasing up to 3.2 GPa

before it slows down and stabilizes above 5.3 GPa. The maximum of the pressure derivative $-dR/dP$ can be used to define the poly-amorphic transition pressure $P_{aa}$[11]. It yields $P_{aa}$ = 3.7 ± 0.4 GPa for GeSe, where the pre-peak intensity is about 20% of that at ambient pressure. We will show later that the $P_{aa}$ of GeTe can be estimated from the kink of the pressure dependence of the first main peak position $Q_1$ of $S(Q)$.

The diffraction patterns of the amorphous GeTe are interfered with by the partial crystallization at and above 3.4 GPa. The analysis of the amorphous phase requires the separation of the contribution of the amorphous and the crystallized states since the total intensity is composed of the diffuse scattering from the amorphous state and the Bragg peaks of the cubic-type crystalline state ($Fm\bar{3}m$). The latter can be accurately fitted by Rietveld refinement, and subtracted to remove the crystalline contribution (see Supplementary Notes and Supplementary Fig. 2b). Figure 3b shows the $G(r)$ from the $I(Q)$ of the amorphous-only contribution. The pressure dependence of $R$ shows a similar error-function shape to that of GeSe (Fig. 3c). It is clear that, comparing to GeSe, the $R$ of GeTe decreases with a steeper slope with increasing pressure, implying a faster vanishing of the Peierls-like distortion and a lower transition pressure $P_{aa}$ in GeTe. This is consistent with the faster decay of the pre-peak intensity of GeTe (Fig. 2c). The significance of the Peierls-like distortion in the polyamorphic

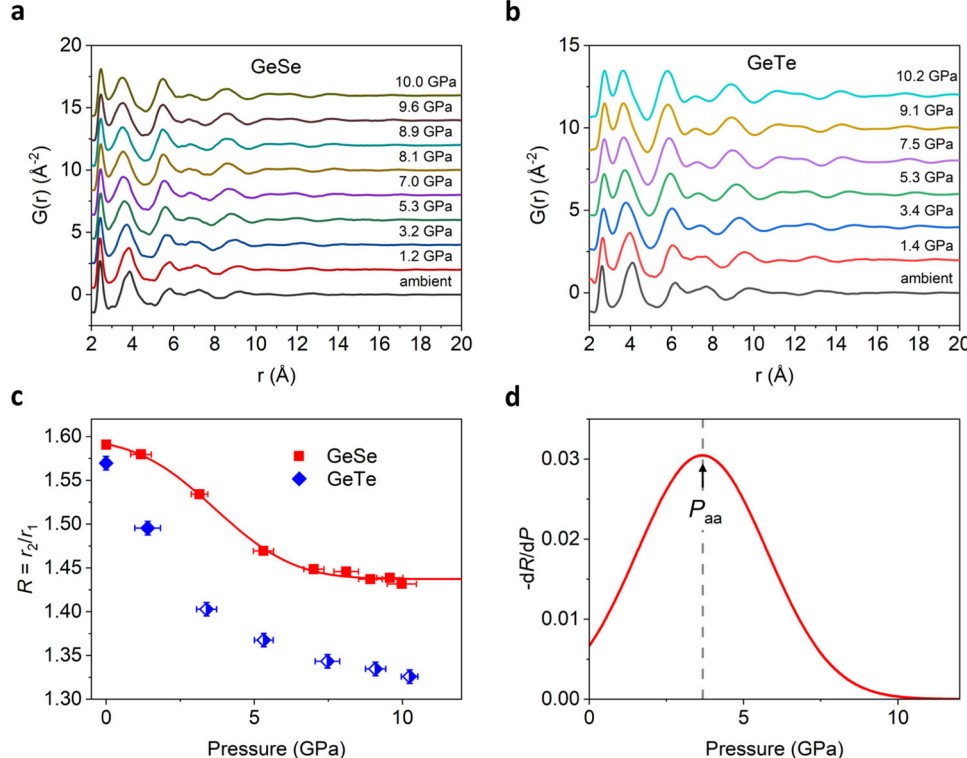

**Fig. 3 | The reduced pair distribution function $G(r)$ from ambient pressure to ~10 GPa. a** GeSe and **b** GeTe. The $G(r)$ of GeTe is obtained after subtracting the crystalline contribution from the diffraction intensity $I(Q)$. The curves are vertically shifted by 2.0 Å$^{-2}$ for clarity. **c** The pressure dependence of the ratio of the second to the first nearest neighbor distance $R = r_2/r_1$ for GeSe (square) and GeTe (rhombus). The peak positions $r_1$ and $r_2$ are determined from the fitting with cubic spline

interpolation. The solid line is the fitting by an error function. The half-filled symbols of GeTe are the data points obtained after removing the contribution of the partial crystallization (see the details in Supplementary Info). **d** The pressure derivative of the fitted $R$ curve for GeSe. The maximum in $-dR/dP$ is used to define the polyamorphic transition pressure $P_{aa} = 3.7$ GPa (dashed line), corresponding to the inflection point of the error function in (**c**).

transition will be discussed in the later section with the atomic-level insight from our NNMD simulations.

The structural evolution of the polyamorphic transition also manifests itself in the pressure dependence of $Q_1$ and the full width of half maximum (FWHM) of the first main diffraction peak of $S(Q)$. As shown in Fig. 4a and b, for both of GeSe and GeTe, the $Q_1$ exhibits a kink and its slope becomes shallower in the high-pressure state. A similar kink has been observed in the polyamorphic transition of Ge$_1$Sb$_2$Te$_4$[20]. By splitting the data points into two subsets, we estimate the transition pressure as the intersection of the two linear fits. As such, the transition pressure $P_{aa}$ of GeSe is estimated as 3.5 ± 1.4 GPa, which agrees well with the one obtained from the error function fitting (3.7 ± 0.4 GPa) in Fig. 3c. The data point at 3.2 GPa around the $P_{aa}$ is grouped into the low-pressure subset in this analysis. The $P_{aa}$ is estimated as 3.7 ± 1.0 GPa if the data point is included in the high-pressure subset, showing that the result is robust to the selection of data points in splitting. With the same method, the $P_{aa}$ of GeTe is estimated to be 1.8 ± 0.4 GPa, which is markedly lower than that of GeSe.

Indeed, the pre-peak intensity of GeSe around the $P_{aa}$ is reduced to about 20% of that at ambient pressure, while for GeTe the same reduction is obtained at the significantly lower pressure of 1.4 GPa. It is worth noting that an earlier study showed a correlation between the temperature-induced LLT and the pre-peak intensity in PCMs, suggesting that the LLT is induced by the onset of Peierls-like distortion during cooling[11]. In this context, our analysis of pre-peak intensity and pair distribution functions suggests that the vanishing of Peierls-like distortion is the underlying microscopic structural change of the polyamorphic transition under compression. We note that the pre-peak of GeTe re-emerges after the pressure release from 10.2 GPa, suggesting the recovery of the Peierls-like distortion (Supplementary

Fig. 4). Given the similar pressure response of GeSe, the reversible behavior of the Peierls-like distortion is also expected for GeSe after the pressure release.

**Compressibility and Coherence length**

The peak position $Q_1$ can be related to the atomic volume ratio in a solid as $\frac{V(P)}{V(P_{amb})} = \left(\frac{Q_1(P_{amb})}{Q_1(P)}\right)^3$ via Ehrenfest's relation $Q_1 = K/d$, where $d$ is the average interatomic spacing corresponding to the position of first diffraction peak, $V$ is the atomic volume, $P$ is the pressure, $P_{amb}$ is the ambient pressure, and $K$ is a constant[30]. The validity of Ehrenfest's relation has been widely confirmed for various amorphous systems such as oxide glasses, bulk metallic glasses, and chalcogenide glasses with good glass forming ability[31,32], although it has not been tested for PCMs. Assuming the validity of the relation in the present systems, the pressure dependence of the atomic volume ratio of GeSe and GeTe can be obtained and shown in the insets of Fig. 4a. The bulk modulus $B$ can be estimated from the linear fit of the volume ratios by using the following equation:

$$\frac{V(P)}{V(P_{amb})} = 1 - \beta(P - P_0) \tag{1}$$

where $\beta = 1/B$ is the compressibility, and $P_0$ is the pressure at a reference point. The ambient pressure is selected as the reference point for the low-pressure states below the transition pressure $P_{aa}$. For the high-pressure states, the lowest pressure point after the polyamorphic transition is used as the reference point.

The bulk modulus $B$ of GeSe is estimated to be 18.8 ± 0.7 GPa (compressibility $\beta = 0.055 ± 0.002$ GPa$^{-1}$) for the low-pressure state. It

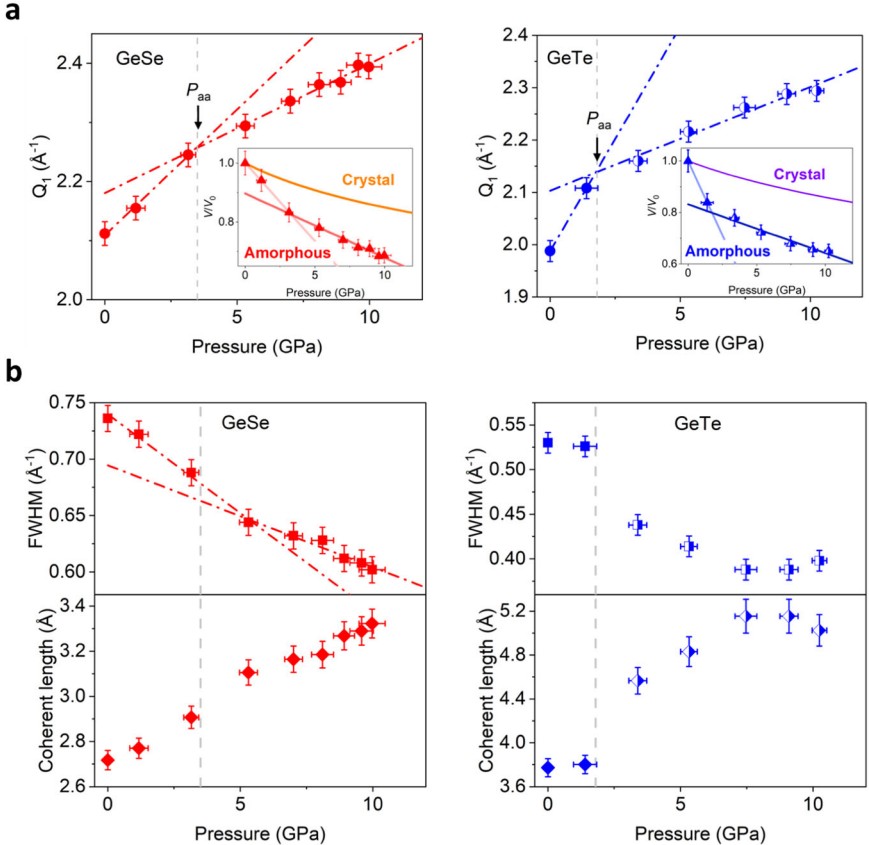

**Fig. 4 | Pressure dependence of parameters of the first diffraction peak of the total structure factor $S(Q)$ for GeSe (left) and GeTe (right). a** Peak position $Q_1$. The dot-dashed lines show the linear fit for the data points for estimating the polyamorphic transition pressure $P_{aa}$. The insets show the pressure dependence of volume ratio to ambient condition calculated from Ehrenfest's relation. The shallower pressure dependence after the polyamorphic transition suggested a significant drop in compressibility (or a rise in the bulk modulus). The orange and purple lines are the volume ratios of the crystalline GeSe and GeTe calculated from the third-order Birch-Murnaghan equation of state with the reported parameters[33]. **b** FWHM of the first diffraction peak (top) and the structure coherent length (bottom). The data points of GeTe with half-filled symbols were obtained from the analysis of estimated amorphous contribution (see Supplementary Fig. 2a).

increases by a factor of 2.4 to $B = 45.9 \pm 2.3$ GPa ($\beta = 0.022 \pm 0.001$ GPa$^{-1}$) for the high-pressure state, indicating lower compressibility after the polyamorphic transition. The $B$ of GeTe is $8.7 \pm 6.8$ GPa ($\beta = 0.12 \pm 0.08$ GPa$^{-1}$) for the low-pressure state, and increases by a factor of 6 to $B = 52.9 \pm 6.9$ GPa ($\beta = 0.019 \pm 0.003$ GPa$^{-1}$) for the high-pressure state. The difference in compressibility between the low- and high-pressure states is reminiscent of that of amorphous Ge$_2$Sb$_2$Te$_5$[19,23]. Xu et al. using ab-initio molecular dynamics simulations, reported that the low-pressure compression of Ge$_2$Sb$_2$Te$_5$ mainly affected the low electron density regions (LED), resulting in a higher compressibility due to the weak van der Waals force over the LED[19]. At higher pressures, most of the LED are squeezed out and the bonds are harder to be compressed[19]. It is reasonable to assume that the amorphous structure of GeTe and GeSe at ambient pressure is spatially heterogeneous with similar LEDs[17]. Then the decrease of compressibility across the transition can be understood with the same mechanism proposed for Ge$_2$Sb$_2$Te$_5$. This is also supported by the similarity of the bulk modulus of the high-pressure amorphous states and the crystalline states of both GeTe and GeSe. The solid lines in the inset of Fig. 4a represent the fitted curves for the data of the crystalline states of GeSe and GeTe reported earlier, using the third-order Birch–Murnaghan equation of state[33]. For GeTe, the high-pressure amorphous state shows a comparable bulk modulus $B = 52.9 \pm 6.9$ GPa ($\beta = 0.019 \pm 0.003$ GPa$^{-1}$) to that of rock-salt GeTe, for which $B = 49.9 \pm 3.2$ GPa ($\beta = 0.020 \pm 0.001$ GPa$^{-1}$). Analogously, the high-pressure state of GeSe also exhibits a comparable bulk modulus $B = 45.9 \pm 2.3$ GPa ($\beta = 0.022 \pm 0.001$ GPa$^{-1}$) to that of the orthorhombic phase of the crystalline GeSe, $B = 40.7 \pm 3.5$ GPa ($\beta = 0.025 \pm 0.002$ GPa$^{-1}$).

Figure 4b shows the FWHM and the coherence length $\xi$ of the amorphous states derived from the FWHM of the first diffraction peak by $\xi = 2/$FWHM[34]. The validity of this inverse relation has been confirmed for bulk metallic glasses, using the coherent length obtained from the Ornstein–Zernicke analysis[31,35]. The FWHM of GeSe exhibits a kink around the transition pressure of 3.5 GPa, whereas the FWHM of GeTe exhibits a distinct drop above that of 1.8 GPa. Since the first peak partially overlaps with the second peak, the FWHM is estimated in three different ways and the results provide qualitatively the same conclusion (see Supplementary Note and Supplementary Fig. 5). Clearly, the coherence length shows a considerable increase after the polyamorphic transition, indicating a higher structural coherence of the high-pressure state for both GeSe and GeTe.

## Molecular dynamics simulations based on a machine-learned neural-network potential

Our experimental results have demonstrated clear evidence for the reversing of the Peierls-like distortion (Figs. 2 and 3) accompanied by the changes in the compressibility and the coherence length upon the polyamorphic transition. To obtain an atomic-level insight, we performed molecular dynamics simulations of amorphous GeTe using a machine-learned neural-network (NN) potential[36], as well as ab initio molecular dynamics simulations of GeSe based on density-functional-theory (DFT) (since no NN potential was available for GeSe). First, we

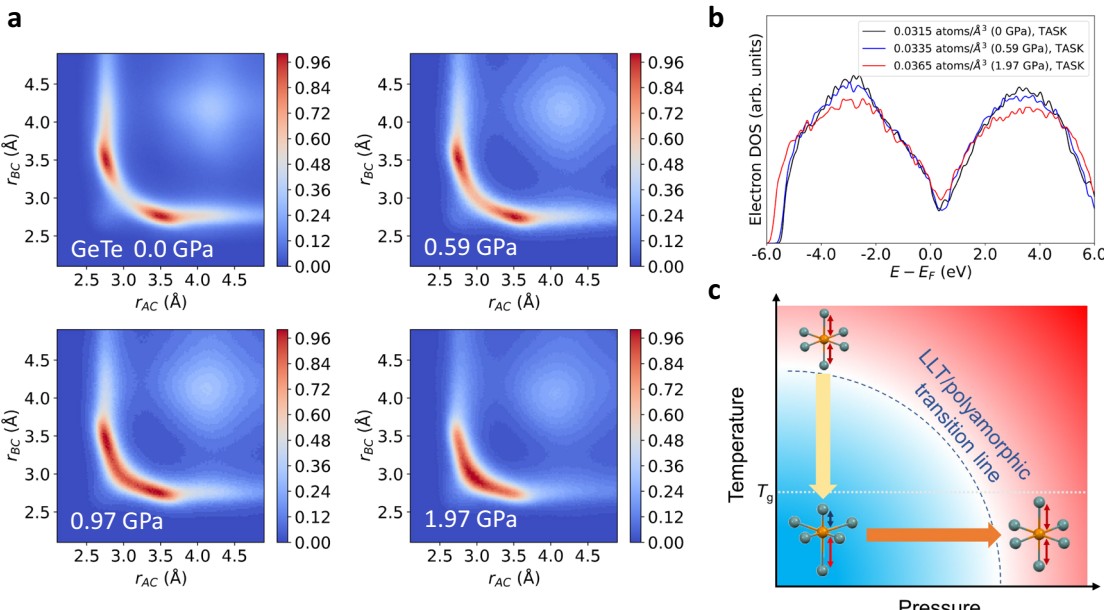

**Fig. 5 | Simulations of amorphous GeTe under pressure. a** Angular limited three-body correlation (ALTBC) plots of GeTe at 0.0, 0.59, 0.97 and 1.97 GPa showing suppression of Peierls-like distortion for increasing pressure. **b** The electronic density of states (DOS) of amorphous GeTe at three representative pressures, 0.0, 0.59 and 1.97 GPa. The DOS exhibits a reduction of the pseudo-bandgap near the Fermi level with increasing pressure. **c** A pressure–temperature ($P$–$T$) schematic illustrating the influence of pressure and temperature on the Peierls-like distortion in $p$-bonded amorphous solids or liquids of PCMs. The dashed line represents specific $P$–$T$ combinations, separating the regions with and without the Peierls-like distortions and corresponding to the possible polyamorphic transitions or LLTs at high temperatures and high pressures. Under this line, numerous $P$–$T$ combinations (shaded blue area) may be used to achieve different levels of Peierls-like distortions for materials property tuning.

discuss GeTe. Models of 512 atoms were constructed with an atomic density of 0.0315, 0.0335, 0.0345 and 0.0365 atoms/Å³ corresponding to the pressure of 0.0, 0.59, 0.97 and 1.97 GPa, respectively. The pair distribution functions $g(r)$ obtained from the models qualitatively reproduced the pressure dependence of experimental $g(r)$ (see Supplementary Figs. 7 and 8). To analyze the Peierls-like distortion in disordered systems such as amorphous materials, it is required to consider groups of three atoms that are "nearly aligned" in the same direction for identifying the alternating long and short bonds. For this purpose, the angular limited three-body correlation (ALTBC) plot has been employed for elucidating the two-dimensional correlation of the interatomic distance of the three atoms that are confined within a certain limiting angle[11].

Figure 5a shows ALTBC plots obtained from the simulation cells at the four pressure conditions of 0.0, 0.59, 0.97, and 1.97 GPa. The limiting angle is set as $\theta_{lim} = 155°$ in the analysis. The labels of axes $r_{AC}$ and $r_{BC}$ are the interatomic distances from one atom to the other two atoms. The plots at 0.0 and 0.59 GPa display two diagonally symmetric peaks around $r_{AC} = 2.7$ Å and $r_{BC} = 3.5$ Å and vice versa. They clearly show that when the bond length on one side is around 2.7 Å, the bond length on the other side tends to be significantly longer by 0.8 Å, which confirms the presence of pairs of long and short bonds and, thus, Peierls-like distortion[11,16,37,38]. With the pressure increase, the two peaks approach each other and merge into a single peak around $r_{AC} = r_{BC} = 2.9$ Å at 1.97 GPa, indicating the suppression of Peierls-like distortion. This result is in excellent agreement with the estimated transition pressure and the vanishing of the pre-peak, as well as with the pressure dependence of $R$ from the experimental data. We stress that our simulations can also reproduce the disappearance of the pre-peak at high pressure, as shown in Supplementary Fig. 9. Furthermore, they provide $S(Q)$ curves in qualitative agreement with experimental ones at different pressures (Supplementary Fig. 10).

As far as GeSe is concerned, we studied 512 atoms models at atomic densities of 0.042 and 0.046 atoms/Å³ corresponding to

pressures of 2.42 and 5.17 GPa. We considered only two pressures due to the high computational cost of ab initio simulations. The resulting ALTBC plots are shown in the supplement (Supplementary Fig. 11). They show that, at 2.42 GPa, there is still pronounced Peierls-like distortion and that the distortion has mostly disappeared at 5.17 GPa. This is in qualitative agreement with a very recent computational paper[39], in which the suppression of the distortions is reported to occur at even higher pressure, possibly due to the absence of van der Waals corrections in the simulations. Thus, a polyamorphic transition occurs in GeSe as well but at higher pressure with respect to GeTe, in full agreement with experimental data.

The $R$ of GeSe and GeTe at ambient pressure is equal to 1.59 and 1.57, respectively, which is close to the $R$-value of the ideal tetrahedrally coordinated system, 1.61. Theoretical and experimental studies have provided evidence for the existence of tetrahedral structural motifs promoted by homopolar Ge–Ge bonds in amorphous GeTe[16,40,41]. They have also found that some tetrahedral structures are present in amorphous GeSe, albeit with a lower concentration than in GeTe[16]. This may suggest another possible explanation for the dependence of $R$ on pressure, namely that the population of tetrahedral motifs changes drastically during compression. Indeed, in our simulations of GeTe the fraction of Ge atoms with tetrahedral coordination decreases from 33.3% to 21.5% upon increasing pressure from 0.59 to 1.97 GPa (see Supplementary Table 1, where zero-pressure data are also shown). To disentangle the two effects, we also investigated amorphous models of pure antimony, which also display Peierls-like distortions but do not have tetrahedral motifs. For this purpose, we performed NNMD simulations at different pressures using a recently developed machine-learned potential for Sb[42]. Similar to GeTe, we observed a suppression of Peierls-like distortions at high pressure and a corresponding reduction in $R$ (see Supplementary Figs. 12 and 13). These findings show unambiguously that changes in Peierls-like distortions significantly affect the value of $R$.

We also performed DFT calculations for GeTe using the TASK functional[43] to identify the evolution of the electronic density of states (DOS) near the Fermi level $E_F$ upon the increase in pressure. Figure 5b shows the DOS of amorphous GeTe at three different pressures of 0.0, 0.59, and 1.97 GPa. The difference in DOS near $E_F$, albeit small, suggests the closing of the pseudo-bandgap with increasing pressure. Similar results are obtained using the computationally more expensive hybrid functional HSE06 (see Supplementary Fig. 14). When the pressure surpasses 2 GPa, GeTe crystallizes rapidly in the simulations, hindering the determination of the DOS of the amorphous phase at higher pressures. The closing of the pseudo-bandgap upon pressure is opposite to the opening of the gap during the metal-to-semiconductor transition in the supercooled liquid state of the same compound[15]. This analysis reveals a correlation between the Peierls-like distortions and the pseudo-bandgap size under high pressures, suggesting a semiconductor-to-metal transition connected to the polyamorphic transition.

## Discussion

The present results demonstrate clear evidence that the elimination of the Peierls-like distortion is the microscopic structural origin of the polyamorphic transition in PCMs under high pressures, which is effectively a reversal of the structural effect observed in concomitance with the metal-semiconductor transition induced by supercooling in PCMs. This argument is supported by both computational and experimental studies. Cobelli et al., using DFT simulations, investigated the local structures and the electronic DOS and Tauc optical bandgap of GeTe and $Ge_2Sb_2Te_5$ in their supercooled liquid state[15]. They estimated a metal-semiconductor transition temperature $T_{M-SC}$ around 800 K for GeTe, about 20% below $T_m$, by identifying the opening of the pseudo-gap in the DOS and of the optical gap, and suggested that the transition is correlated with the enhancement of Peierls-like distortion. Their observation also suggested that the transition is of higher-order (continuous) character rather than a first-order transition, which is in line with the recent experimental study of liquid tellurium[15,44]. A femtosecond X-ray diffraction experiment by Zalden et al. revealed evidence of Peierls-like distortion associated with an LLT and a metal–semiconductor transition during deep supercooling of liquid $Ge_{15}Sb_{85}$ and $Ag_4In_3Sb_{67}Te_{26}$ while avoiding crystallization[11]. We expect that pressure should have similar effects on the Peierls-like distortions in the amorphous state of these two PCMs as well. This is because under high pressure the longer bonds are more easily shortened than the shorter bonds are elongated. This also explains why the changes in $R$ under pressure are mainly due to the decrease of $r_2$, whereas the change of $r_1$ is small. The symmetric structures lead to smaller atomic volumes than the Peierls-like distortions[45]. Although it has been debated whether the polyamorphic transition is necessarily related to an LLT[2], it appears that in PCMs they share the same underlying mechanism. If an LLT occurs in the supercooled liquid at the ambient pressure, and a polyamorphic transition occurs at room temperature, we expect a possible LLT line of certain pressure-temperature combinations in the pressure-temperature metastable phase diagram, as discussed in other types of anomalous liquids[2,14]. Identifying this line would require challenging experiments combining a high-pressure environment with pump-probe femtosecond X-ray diffractions to avoid fast crystallization in the supercooled liquid under high pressures. Figure 5c shows a schematic of the change of Peierls-like distortions in the metastable pressure–temperature (P–T) diagram for liquid and amorphous solid states of p-bonded PCMs. The dashed line separates the regions with and without Peierls-like distortions, corresponding to the line of a (likely continuous) LLT or polyamorphic transition. An interesting implication of this P–T diagram is the presence of numerous P–T combinations within the triangle-like region (blue-shadowed area) that can be employed to achieve varying degrees of Peierls-like distortions and fine-tune materials properties.

Although GeTe and GeSe are both isoelectronic (predominantly) p-bonded systems, the levels of distortion of local octahedral-like atomic arrangement are different. In their crystalline phase, GeTe is characterized by a small degree of Peierls distortion deviating from a perfect octahedral arrangement, while GeSe has a larger degree of Peierls distortion[46]. This difference manifests itself in the bonding mechanisms, which can be characterized by the number of electrons shared between neighboring atoms. GeTe has a markedly lower number of shared electrons (less covalent) compared to GeSe, which behaves like an ordinary covalently bonded system[47]. This could explain many property differences between GeTe and GeSe such as optical contrast, Born effective charge, and the Grüneisen parameter for transverse optical modes[47]. In the amorphous phase, GeSe is also considered more covalent than GeTe, as reflected by the larger Peierls-like distortion. The higher degree of rigidity and covalency in GeSe is thought as the origin of lower strength of secondary (β-) relaxation and lower electronic conductivities in glassy GeSe[37,48]. In this light, it is plausible that reversing the Peierls-like distortion necessitates a higher pressure for GeSe compared to GeTe. This could account for GeSe exhibiting a higher polyamorphic transition pressure than GeTe.

The different behavior in the crystallization between GeTe and GeSe can be understood by considering the different levels of Peierls-like distortion in the amorphous state with respect to the structure of the crystalline phases. The vanishing of the pre-peak of GeTe at 3.4 GPa suggests the suppression of alternating long and short bonds at this pressure. This implies that the local structural motifs of the amorphous state become similar to the rock-salt structure of the crystalline state with the near-ideal octahedral coordination[11]. This results in a smaller interfacial energy between the two states, lowering the energy barrier for the nucleation and triggering the partial crystallization at that pressure. By contrast, the amorphous GeSe exhibits slower decay of the pre-peak of $I(Q)$ up to 9.1 GPa, suggesting the suppression of the Peierls-like distortion around 10 GPa. As a comparison, crystalline GeSe exhibits the GeS-type orthorhombic phase with distorted octahedral coordination up to 82 GPa[33]. Thus, the larger dissimilarity between amorphous and crystalline structures of GeSe persists to much higher pressures, leading to higher interfacial energy to overcome the nucleation of crystals in GeSe under compression. In a recent study of pseudo-binary GeTe–GeSe alloys, Persch et al. showed that the crystallization kinetics of GeSe is several orders of magnitude slower than that of GeTe upon laser heating at ambient pressure. The authors correlated this difference with the higher covalency of GeSe and speculated lower interfacial energy between liquid and crystal in GeTe[24].

We have demonstrated that the pressure-induced reversal of the Peierls-like distortion in amorphous GeSe and GeTe elicits a polyamorphic transition. The high-pressure amorphous state is characterized by a distinct rise in the bulk modulus (i.e. lower compressibility), a larger coherent length, and the vanishing of Peierls-like distortions. The experimental observations are supported by the ALTBC analysis of the NNMD simulations. The DFT calculations revealed an increase in the electronic density of states with pressure at the Fermi level, which suggests a possible semiconductor-to-metal transition. The mechanism of polyamorphic transition appears the same as that underlying the temperature-induced LLT in the supercooled liquid of PCMs. This implies a possible LLT line of certain pressure-temperature combinations in the P–T metastable phase diagram. Identifying this line at high temperature and pressure would be of future interest not only for PCM research but also in the broader context of anomalous liquids (e.g. tellurium and water). A previous study of the melts of GeSe and GeTe noticed a kink in the pressure dependence of the structural parameters around 1–3 GPa above $T_m$[29]. Yet, how (if at all) the kink is related to the present polyamorphic transition requires further studies to clarify. Lastly, applying pressure, like temperature, appears to be an effective way to tune the Peierls-like

distortion and thus alter the properties of the PCM. The effectiveness of pressure may vary with the level of Peierls-like distortion at the ambient pressure, depending on the components and nature of bonding in the $p$-bonded systems, as evidenced by the cases of GeTe versus GeSe. This insight might be potentially useful for engineering desirable material properties.

## Methods

### In-situ high-pressure pair distribution function measurement

Amorphous GeSe and GeTe samples were prepared by dc magnetron sputtering depositions (base pressure ~$10^{-6}$ mbar and argon flow of 20 sccm) from stoichiometric targets with purity higher than 99.99%. As-deposited films with a thickness of 5–7 μm were scraped off from the substrates and cut into small flakes for X-ray scattering experiments. In-situ high-pressure pair distribution function measurement was performed by using Paris–Edinburgh (PE) press[26] at the BL05XU beamline in SPring-8. The PE press with 120 degrees of aperture in the horizontal direction is suitable for measuring diffraction data in a wide range of momentum transfer $Q$. High-pressure experiment was conducted by using cupped-Drickamer-Toroidal (CDT)-type anvil cell[49], which consists of a polycarbonate ring, a boron-epoxy gasket, a MgO ring, and Cr-doped MgO caps. A sample pellet with a diameter of 1.3 mm and a height of 1.0 mm was placed into the MgO ring with BN caps. Pressures were determined by the equation of state of Au[50], which is placed on a side of the sample.

Pair distribution function measurements were conducted by using a high-flux pink beam at the photon energy of an X-ray beam of 100.1298 keV, which was generated with a double multilayer monochromator at the BL05XU[51]. Diffraction data were collected by scanning 2$\theta$ angle by using two-point detectors up to 31.8°, corresponding to $Q = 27$ Å$^{-1}$. The measured scattering data $I(Q)$ were transformed into the total structure factor $S(Q)$ and reduced pair distribution function $G(r)$. The details of the data processing are shown in the Supplementary Notes and Supplementary Figs. 1–3. The program Synchrotron Powder and Plotpro was used for the extraction of amorphous contribution from the scattering data by using the Split-type Pearson-VII functions[52]. The background subtraction, correction, and normalization were applied for converting the $I(Q)$ into the total structure factor $S(Q)$ and the reduced pair distribution function $G(r)$ on the software pdfgetX2[53]. The positions of peaks in $S(Q)$ and $G(r)$ are estimated by the cubic spline fitting.

### Molecular dynamics and DFT simulations

In this work, the MD simulations for GeTe were conducted with the MD driver DL_POLY[54] combined with an NN potential for GeTe (RuNNer[55]), which was successfully employed to study the structural and kinetic properties of liquid and amorphous GeTe[56]. We considered four models with 512 atoms in a cubic simulation box. The atomic densities of the four models were 0.0315, 0.0335, 0.0345, and 0.0365 atoms/Å$^3$, respectively. The amorphous phases were generated by a melt-quench process. The models were first heated to above 2500 K for full randomization. Then the systems were equilibrated at 1000 K for 200 ps, after which they were quenched to 300 K with a quenching rate of 5 K/ps[57]. Finally, the systems were annealed at room temperature for 1 ns. At each density, the ALTBC was calculated with 4000 samplings in the simulation time range of 0.8 ns with a sampling interval of 1 ps. NVT simulations were performed using a stochastic thermostat[58] and the time step was set to 0.002 ps.

The MD simulations for GeSe were carried out using the CP2K package[59]. We employed the second-generation Car-Parrinello method[60] implemented in the QUICKSTEP module of this package. A double-zeta plus polarization Gaussian-type basis set was used to expand the Kohn–Sham orbitals and plane waves with a cutoff of 300 Ry were used to expand the charge density. We employed the same function used for the generation of the NN potential for GeTe[56].

We considered two models with 512 atoms in a cubic simulation box at atomic density 0.042 and 0.046 atoms/Å$^3$, respectively. The amorphous phases were generated with a similar melt-quench process as for GeTe, albeit with shorter quenching and annealing times. The models were first heated to above 2500 K for full randomization, then they were equilibrated at around 1000 K for 30 ps, after which they were quenched to 300 K with a quenching rate of 15 K/ps and annealed at room temperature for 30 ps. The time step was set to 0.002 ps.

The DFT simulations for GeTe were also performed using QUICKSTEP. In this case, a triple-zeta plus polarization Gaussian-type basis set and plane waves with a cutoff of 300 Ry were used to expand the Kohn-Sham orbitals and the charge density, respectively. Models containing 512 atoms were generated by NNMD simulations; subsequently, their geometry was optimized with a conjugate gradient method implemented by the authors in the DL_POLY package. The electronic density of state (DOS) was computed using the TASK exchange[43] and the PW92-LDA correlation[61]. The DOS curve was obtained using a Gaussian broadening with a width of 0.1 eV.

## Data availability

All data is available in the manuscript and the Supplementary Information. Source data are provided with this paper.

## Code availability

Custom code used in this study is available from the corresponding authors upon request.

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

## Acknowledgements

The synchrotron radiation experiment was performed at BL05XU of SPring-8. This work was supported by a research grant (42116) from VILLUM FONDEN (S.W.). We thank the Danish Agency for Science, Technology, and Innovation for funding the instrument center DanScatt. This work was financially supported by the Japan Society for the Promotion of Science (JSPS) KAKENHI Grant No. JP19KK0132 (E.N.), JP20H00201 (Y.K.), JP21H05235 (E.N.), and JP21J12479 (T.F.). This research is supported by the SACLA/SPring-8 Basic Development Program (Y.K.). Y.C. acknowledges the financial support of the China Scholarship Council.

## Author contributions

S.W. and R.M. jointly supervised this work. S.W., E.N., and Y.K. initiated the project. T.F., Y.K., E.N., S.T., H.K., K.O., H.Y. (Hirokatsu Yumoto), T.K., H.Y. (Hiroshi Yamazaki), Y.S., H.O., I.I., Y.H., and M.Y. contributed to the experimental methods and performed the experiment with samples from S.W. T.F. and S.W. analyzed experimental data. Y.C., R.M., D.C., and M.B. developed the computational methods and performed NNMD simulations. T.F., S.W., R.M., and Y.C. wrote the manuscript with important input from Y.K., E.N., M.B., H.K., and others.

## Competing interests

The authors declare no competing interests.
