## [Peer Review File · Nature Communications]

REVIEWER COMMENTS

Reviewer #1 (Remarks to the Author):

In this manuscript, the authors combined experimental and theoretical simulations to reveal the mechanism of polyamorphism in chalcogenide glasses under pressure, demonstrating that the reversal of pressure-induced Peierls-like distortions is the primary origin of this glass-glass transition. The results of this article are significant for understanding the mechanism of phase change materials and designing new materials. The work is solid and well-written, and thus I recommend the publication after minor revision.

1. In Figure 3, the author demonstrates the evolution of the pair distribution function under pressure, while calculating $R = r_2 / r_1$ to prove the disappearance of Peierls-like distortion. However, I noticed that the change in R under pressure is mainly due to the decrease of r_2 , while r_1 hardly changes. This phenomenon is quite interesting, and I hope the author can discuss more about the reason behind it, which can also help readers understand the mechanism of the amorphous structure change under pressure.

2. In Figure 3c, the authors calculated the curves of R as a function of pressure for GeTe and GeSe, reflecting the weakening of Peierls-like distortion with increasing pressure. Then in Figure 3d, by calculating the derivative of the R curve with respect to pressure, the P_{aa} of GeSe was determined. However, the data for GeTe is not shown in the figure, and it is hoped that the authors can provide it. Similarly, in Figure 5b, the authors should show the DOS data without pressure in order to compare the changes in pseudo-bandgap.

3. In this study, a molecular dynamics method based on a machine learning potential function was utilized to investigate the structural transformation of amorphous GeTe under pressure, which is a popular approach and has important implications for studying large-scale amorphous materials systems. However, it should be noted that although machine learning potential functions are more efficient, their computational accuracy is generally lower than first-principles methods, so caution should be exercised when using them. Here, it is hoped that the authors can explain the significance or advantages of using a machine learning potential function in this work, as well as provide detailed information on the training method and accuracy of the potential function.

4. In this article, the authors demonstrated the evolution of Peierls-like distortion and polyamorphic transition under pressure using experimental and theoretical methods, which is of great significance for understanding the mechanism of PCMs and designing new materials. However, what will happen to the structure of the amorphous material if the pressure is released before GeTe crystallization (for example

2.0 GPa)? Can the phase of this polyamorphic transition exist stably? It is hoped that the authors can discuss this issue.

5. In the text, it is stated that "Indeed, the fraction of Ge atoms with tetrahedral coordination decreases from 33.3% to 21.5% upon increasing pressure from 0.59 GPa to 1.97 GPa." Is this conclusion drawn by the authors through experiments or simulations? If not, please give the citation.

Reviewer #2 (Remarks to the Author):

The manuscript of Fujita et al. reports the pressure-induced polyamorphic transition in GeSe and GeTe, and provide evidence that the suppression of Peierls-like distortions elicits the transition. Well-designed study and the work will be of significance to the field and related community. I recommend it for publication after some minor issues listed below have been taken care of.

1. Line 79: X-ray beam energy of 100.1298 keV, why 4 decimal digits? And in line 99, it's stated as 100 keV (no decimal digit now)
2. Did authors check if GeSe turns back to its low pressure phase after pressure release?
3. Line 133: I think the unit of the alpha parameter should be "GPa".
4. Line 272: How did the authors get these atomic number density values? Are these measured values? or picked from the literature?
5. Line 426: How about background correction? Maybe better to plot all the corrections applied along with the raw data and normalization etc?
6. Any idea about the origin of the first nearest neighbor peak (located at r_1) and the second one (located at r_2)? Partial pair distribution functions might help on this.
7. Any discussion about the effect of inhomogeneity of sample environment on these results since this is not a diamond anvil cell experiments including Ne or He inert gases as the pressure transmitting media?

Reviewer #3 (Remarks to the Author):

In the manuscript “Pressure-induced reversal of Peierls-like distortions elicits the polyamorphic transition in GeTe and GeSe”, the authors report the underlying mechanism of solid-state transition induced by pressure in the amorphous phase of GeTe and GeSe chalcogenide compounds. Using high-energy synchrotron X-rays, the authors manifested the effect of pressure on the amorphous structures, which proves that pressure can reverse the Peierls-like distortions. Further, the polyamorphic transition pressure P_{aa} is 3.7 GPa for GeSe and 1.8 GPa for GeTe, respectively. Moreover, combining with density functional theory and molecular dynamics simulations, the authors revealed the change of local motifs and electronic structures in the GeTe amorphous under pressure using different atomic density GeTe amorphous models. This work is publishable after the authors have addressed the following comments and suggestions.

1. The authors employed molecular dynamics simulations to generate GeTe amorphous models, more details for generating the amorphous models are therefore necessary to provide. The authors are suggested to reference a piece of previous work (J. Mater. Chem. C 11 (2023) 1360-1368) on amorphous phase change materials which provide useful information for generating amorphous models.

2. The calculated pair distribution function $g(r)$ of GeTe amorphous in Fig. S4 is different with the $g(r)$ obtained by experiment. The intensity of the second nearest neighbor peak in the simulation is weaker, manifesting that the amorphous GeTe in simulation is more disordered than that in experiments. The authors should carefully check the GeTe amorphous models, which will influence the results of electronic structure. Moreover, it is easy to obtain the $S(Q)$ by molecular dynamics frames. It might be more convincing to compare the $S(Q)$ of experiments and calculations.

3. It seems that the pseudo-bandgap of GeTe amorphous models is caused by the functionals. Using a better functional, e.g., HSE06 or PBE0, the author could obtain more precise electronic structures, which is closer to the situation in experiments.

Reviewer #4 (Remarks to the Author):

In the manuscript entitled “Pressure-induced reversal of Peierls-like distortions elicits the polyamorphic transition in GeTe and GeSe”, Fujita et al. have studied a polyamorphic transition in GeSe and GeTe through high pressure synchrotron X-ray experiments and molecular dynamic simulations. The key focus of the study is to investigate if modulations in Peierls-like distortions elicit a polyamorphic transition in GeSe and GeTe and if so, how they can be tuned to adjust the transition. Having carefully evaluated the manuscript, I am unfortunately unable to recommend it for publication in Nat. Commun. as there are many concerns I have about the manuscript which are listed below, do not go with the high standard of Nat. Commun.

1. How do the authors justify calling the high-pressure phase a distinct amorphous phase? Especially for GeTe, isn't it possible that beyond the so called P_{aa} , the phase is macroscopically amorphous but with microscopic crystallization. All the characterizations provided, viz. reduced Peierls distortion, higher

coherence length, reduced compressibility etc. would be observed even if a material is undergoing a gradual transition from an amorphous to crystalline phase. Also how did the authors estimate the partial crystallization and quantify it? In addition, the authors must study the extent of crystallization at different pressures and see the change in trend near the proposed polyamorphic transition temperatures.

2. The authors should provide why pressure acts to reduce the Peierls-like distortion in the studied systems which is the centric part of this manuscript. Unfortunately, no such discussions have been in the present work.

3. The motivation of the manuscript is very unclear and confusing. Well, in many places in the manuscript, it is reflected that the polyamorphic transition is an outcome of the suppressing Peierls-like distortion. What is the chemical insight of it? Why Peierls-like distortion is responsible for such a phenomenon?

4. Assuming that there is indeed a polyamorphic transition, how are the authors confidently stating that these transitions are 'driven' by the vanishing of Peierls-like distortion? The reduction could be just an effect of pressure on an amorphous phase. For instance, during LLT, if one observes a change in viscosity, one can't attribute it to be the driving force for the LLT.

5. Authors have stated that "With increasing pressure, the sharp Bragg peaks of GeTe appear at 3.4 GPa, indicating partial crystallization, while GeSe remains fully amorphous up to 10.0 GPa." How do the authors evaluate sharpness? I can also see sharpening of the $Q(\text{\AA}^{-1}) \sim 2.4$ peak in Fig. 1a up to some extent at 10 GPa. How did they confirm the amorphous to crystal transition just based on this?

6. Since $S(Q)$ and $g(r)$ are related via a Fourier transform, can the authors confirm if their simulations can reproduce the pre-peak feature and its vanishing with pressure?

7. In Fig. 2c, what are the physical significances of the fitting parameters α and γ ? It is quite clear from the figure itself that the decay is much faster in GeTe than GeSe, therefore there's no need of the fitting the data unless the fitting parameters provide some valuable relevant information.

8. What do the peaks correspond to in $I(Q)$ plots shown in Fig.1. In Fig. S4 (a), why is the $g(r)$ intensity decreasing with increasing pressure? Also, in Fig 3. (a) and (b), the number on y-axis is meaningless since it's a stacked plot.

9. It is really surprising that the authors haven't provided any molecular dynamic simulation studies for GeSe.

10. Since GeSe exhibits strong covalent bonding environment and the bond strength in GeTe is lower compared to GeSe, shouldn't the bulk modulus be lesser in value for GeTe than GeSe?

Response Letter

We thank all four Referees for their review reports and helpful suggestions. Three reviewers (Reviewer #1, 2, 3) give overall positive evaluations; Reviewer #4 has some points of concern which we will address with detailed clarifications. We provide our point-by-point responses to all the comments of the Reviewers sequentially below, citing the reports in *italics*. The changes made in the revised manuscript and SI are marked with blue font color.

Response to Reviewer 1:

In this manuscript, the authors combined experimental and theoretical simulations to reveal the mechanism of polyamorphism in chalcogenide glasses under pressure, demonstrating that the reversal of pressure-induced Peierls-like distortions is the primary origin of this glass-glass transition. The results of this article are significant for understanding the mechanism of phase change materials and designing new materials. The work is solid and well-written, and thus I recommend the publication after minor revision.

1. In Figure 3, the author demonstrates the evolution of the pair distribution function under pressure, while calculating $R = r_2/r_1$ to prove the disappearance of Peierls-like distortion. However, I noticed that the change in R under pressure is mainly due to the decrease of r_2 , while r_1 hardly changes. This phenomenon is quite interesting, and I hope the author can discuss more about the reason behind it, which can also help readers understand the mechanism of the amorphous structure change under pressure.

Response

Typically, in Peierls-distorted systems, the longer bonds are more easily shortened than the shorter bonds are elongated. This phenomenon was observed and discussed in previous work, such as Otjacques et al., PRL 103, 245901 (2009). It is responsible for the negative thermal expansion observed in tellurium-based liquid alloys, in which Peierls distortions disappear at high temperature.

Below we show the nearest-neighbor histograms for the atoms in GeTe at two pressures (0.59 and 1.97 GPa). These histograms indicate that, upon increasing pressure, the 3 peaks corresponding to short bonds hardly change, whereas the 3 broader peaks corresponding to long atomic contacts become more confined and shift to lower distances.

We have added a short discussion about this phenomenon in the main text (Page 10, bottom paragraph).

Figure R1. The nearest-neighbor histograms for the atoms in GeTe at two pressures (0.59 and 1.97 GPa). The 3 peaks corresponding to short bonds hardly change upon increasing pressure, whereas the 3 broader peaks corresponding to long atomic contacts become more confined and shift to lower distances.

2. In Figure 3c, the authors calculated the curves of R as a function of pressure for GeTe and GeSe, reflecting the weakening of Peierls-like distortion with increasing pressure. Then in Figure 3d, by calculating the derivative of the R curve with respect to pressure, the P_{aa} of GeSe was determined. However, the data for GeTe is not shown in the figure, and it is hoped that the authors can provide it. Similarly, in Figure 5b, the authors should show the DOS data without pressure in order to compare the changes in pseudo-bandgap.

Response

For GeSe, the derivative of R with respect to pressure is calculated from the fitted curve of an error function (red curve in Fig.3c). However, for GeTe, the R drops rapidly with increasing pressure; therefore, a plateau of R at low pressures is not obvious, possibly due to the limited number of data points. For this reason, the fit with an error function is difficult for GeTe and P_{aa} cannot be obtained using the derivative of R in this case. Thus, an alternative method is used to estimate P_{aa} of GeTe based on the analysis of Q_1 of $S(Q)$, as demonstrated in Fig.4.

We have added the following sentence in the manuscript to clarify this point (Page 5, the first paragraph):

“The pressure dependence of R of GeSe can be fitted by an error function, while the fit is difficult for GeTe due to the limited number of data points at low pressure (Figure 3c).”

In addition, we have added the DOS data at the zero pressure in Fig.5b for a clear comparison of the changes in the pseudo-bandgap with pressure.

3. In this study, a molecular dynamics method based on a machine learning potential function was utilized to investigate the structural transformation of amorphous GeTe under pressure, which is a popular approach and has important implications for studying large-scale amorphous materials systems. However, it should be noted that although machine learning potential functions are more efficient, their computational accuracy is generally lower than first-principles methods, so caution should be exercised when using them. Here, it is hoped that the authors can explain the significance or advantages of using a machine learning potential function in this work, as well as provide detailed information on the training method and accuracy of the potential function.

Response

The machine-learning potential we used was thoroughly tested in previous work. The first version of the potential was published in 2012 and was shown to provide an accurate description of the structural and kinetic properties of amorphous and supercooled liquid GeTe (Refs. 34 & 37, Sosso et al., Phys. Status solidi **249**, 1880 (2012); Sosso et al., J. Phys. Chem. Lett. **4**, 4241 (2013)). We use a refined version of the potential, which was generated in 2017 (Ref. 48): besides being as accurate as the previous version in describing the bulk properties, it also describes satisfactorily the behavior of surfaces of crystalline and amorphous GeTe. Regarding the validation of the NN potential under pressure, the figure below (taken from Gabriele Sosso's PhD thesis) shows the reference DFT energies versus the predicted NN energies for the whole dataset (training and test points as well). The agreement between the two sets of energies is very good.

Figure R2. Comparison of NN and DFT energies for the whole data sets (training and test points).

4. In this article, the authors demonstrated the evolution of Peierls-like distortion and polyamorphic transition under pressure using experimental and theoretical methods, which is of great significance for understanding the mechanism of PCMs and designing new materials. However, what will happen to the structure of the amorphous material if the pressure is released before GeTe crystallization (for example 2.0 GPa)? Can the phase of this polyamorphic transition exist stably? It is hoped that the authors can discuss this issue.

Response

This is an interesting point. Although we do not have the data for the case where pressure is released *before* the partial crystallization (e.g. 2 GPa), we do have the data for the amorphous phase after releasing the pressure from 10.2 GPa. Figure R3 shows the low-Q range of diffraction intensity of GeTe at ambient pressure before compression, at 10.2 GPa, and at ambient pressure after pressure release. At least 75 % of the original volume of GeTe remained in an amorphous state after the partial crystallization. The existence of the pre-peak after the pressure release demonstrates the recovery of the low-pressure state with the Peierls-like distortion even after the partial crystallization and the compression up to 10.2 GPa. This indicates that the polyamorphic transition is reversible, and the low-pressure phase can stably exist.

In addition, we observed that the partially crystallized cubic GeTe is transformed into the rhombohedral phase after the pressure release. Importantly, the rhombohedral GeTe exhibits “Peierls-distortion” (i.e. the crystalline counterpart of Peierls-like distortion with long-range order of alternating long and short bond sequence). A rhombohedral-to-cubic phase transition induced by pressure was previously reported around 3.0 GPa (Onodera et al., *Phys. Rev. B*, 1997, 59, 7935-7941). This transition pressure, which is close to our polyamorphic transition pressure, suggests that both amorphous and crystalline states favor the Peierls-distorted structure below 3 GPa. Based on this analysis, it is highly likely that the low-pressure amorphous state with Peierls-like distortion will be recovered even when pressure is released before crystallization.

We have added a brief discussion regarding this point in the main text:

“We note that the pre-peak of GeTe re-emerges after the pressure release from 10.2 GPa, suggesting the recovery of the Peierls-like distortion (Figure S4). Given the similar pressure response of GeSe, reversible behavior of the Peierls-like distortion is also expected in GeSe after the pressure release.”

We have also added the following figure in the Supplementary Information.

Figure R3. The diffraction intensity $I(Q)$ of amorphous GeTe at ambient pressure before compression (black), at 10.2 GPa (red), and at ambient pressure after pressure release from 10.2 GPa (orange). The low- Q range was magnified to enhance the pre-peak. The existence of the pre-peak after pressure release indicates the recovery of Peierls-like distortion and indicates the reversibility of the polyamorphic transition.

5. In the text, it is stated that “Indeed, the fraction of Ge atoms with tetrahedral coordination decreases from 33.3% to 21.5% upon increasing pressure from 0.59 GPa to 1.97 GPa.” Is this conclusion drawn by the authors through experiments or simulations? If not, please give the citation.

Response

Thank you for pointing this out. This conclusion is drawn from the simulations. We have rephrased the sentence for better clarity. Furthermore, we have added a table in the supplement (Table S1) showing the concentration of tetrahedra as a function of pressure. In this table, we have also included the data for the zero-pressure model.

Response to Reviewer #2:

The manuscript of Fujita et al. reports the pressure-induced polyamorphic transition in GeSe and GeTe, and provide evidence that the suppression of Peierls-like distortions elicits the transition. Well-designed study and the work will be of significance to the field and related community. I recommend it for publication after some minor issues listed below have been taken care of.

1. Line 79: X-ray beam energy of 100.1298 keV, why 4 decimal digits? And in line 99, it's stated as 100 keV (no decimal digit now)

Response

Thank you for pointing this out. The wavelength of the X-ray beam was calibrated by using the positions of the Bragg peaks of standard samples of CeO_2 , yielding the photon energy of

100.1298 keV. We corrected the digit of the photon energies to 4 decimal digits, which were used to convert the diffraction angle into the wavevector transfer Q .

2. Did authors check if GeSe turns back to its low-pressure phase after pressure release?

Response

Unfortunately, we didn't measure the diffraction data of GeSe after pressure release. On the other hand, we observed the recovery of the low-pressure state in amorphous GeTe (Please see our response to the 4th question of Reviewer #1). Since amorphous GeSe and GeTe follow the same mechanism of polyamorphic transition and exhibit similar pressure response, we expect that amorphous GeSe would also turn back to the low-pressure state after pressure release.

We have added a brief discussion about this point in the revised manuscript.

3. Line 133: I think the unit of the alpha parameter should be "Gpa".

Response

Thank you for pointing out the error. It is now corrected.

4. Line 272: How did the authors get these atomic number density values? Are these measured values? Or picked from the literature?

Response

These density values were chosen in order to build models with varying pressure up to 2 GPa. Although the neural-network potential seems to be accurate even at higher densities (see reply to point 3 by Referee 1 and relevant plot therein), we did not consider pressures above 2 GPa to be on the safe side.

In the new version of the Supplementary Information, we have included additional data (fraction of tetrahedral structures, $S(Q)$) for a fourth model at zero pressure (0.0315 atoms/Å³) to address some of the concerns by the reviewers.

Furthermore, we have included additional ab initio molecular dynamics simulations for GeSe to address point 9 by referee 4. For this system, we have chosen only two density/pressure values due to the high computational cost of DFT simulations. The results (Figure S11) unambiguously show that the polyamorphic transition occurs at higher pressure for GeSe, in agreement with experiments.

5. Line 426: How about background correction? Maybe better to plot all the corrections applied along with the raw data and normalization etc?

Response

We have added the step-by-step explanation of our data processing to the Supplementary Information. Figure S1 shows the flowchart to convert the raw diffraction data $I(Q)$ to the

total structure factor $S(Q)$ and reduced pair distribution function $G(r)$. The results of the corrections and normalization are plotted in Figure S3.

6. Any idea about the origin of the first nearest neighbor peak (located at r_1) and the second one (located at r_2)? Partial pair distribution functions might help on this.

Response

Below we show the partial pair distribution functions at two pressures (0.59 and 1.97 GPa). The first peak is due to Ge-Te bonds and, to a lesser extent, Ge-Ge bonds, whereas the second peak is mainly due to Te-Te contacts.

Figure R4. Partial pair correlation functions of amorphous GeTe at pressure 0.59 GPa (density $0.0335 \text{ atoms}/\text{\AA}^3$) and 1.97 GPa ($0.0365 \text{ atoms}/\text{\AA}^3$). The first peak is due to Ge-Te bonds and, to a lesser extent, Ge-Ge bonds, whereas the second peak is mainly due to Te-Te contacts. All $g(r)$ curves are normalized such that $g(r)$ approaches 1.0 at large r values.

We have included this figure in the Supplementary Information as Figure S8.

7. Any discussion about the effect of inhomogeneity of sample environment on these results since this is not a diamond anvil cell experiments including Ne or He inert gases as the pressure transmitting media?

Response

We added a discussion paragraph about sample environment in Supplementary Information. Figure R5 shows the pressure dependence of the atomic volume ratio V/V_0 of the partially crystallized GeTe obtained in the present study compared with those reported in previous

study, which is measured under hydrostatic environment (Onodera et al., *Phys. Rev. B*, 1997, 59, 7935-7941). Our obtained data are well consistent with the previous study up to 10.2 GPa, indicating that the observed pressure response should be almost the same as the one that would have been observed under hydrostatic conditions. Since we used soft hexagonal BN and MgO as a pressure medium surrounding the sample, the pressure medium would produce near-hydrostatic environment in the Paris-Edinburgh cell.

We have added this discussion and the following figure in the Supplementary Information under the subsection “*The pressure environment of the Paris-Edinburgh press*”.

Figure R5. The pressure dependence of the atomic volume ratio of cubic GeTe obtained from the partially crystallized data. The pressure response of the crystalline phase is consistent with the result of the third-order Birch-Murnaghan fit obtained under hydrostatic conditions (Onodera et al., *Phys. Rev. B*, 1997, 59, 7935-7941).

Response to Reviewer #3:

In the manuscript “Pressure-induced reversal of Peierls-like distortions elicits the polyamorphic transition in GeTe and GeSe”, the authors report the underlying mechanism of solid-state transition induced by pressure in the amorphous phase of GeTe and GeSe chalcogenide compounds. Using high-energy synchrotron X-rays, the authors manifested the effect of pressure on the amorphous structures, which proves that pressure can reverse the Peierls-like distortions. Further, the polyamorphic transition pressure P_{aa} is 3.7 GPa for GeSe and 1.8 GPa for GeTe, respectively. Moreover, combining with density functional theory and molecular dynamics simulations, the authors revealed the change of local motifs and electronic structures in the GeTe amorphous under pressure using different atomic density GeTe amorphous models. This work is publishable after the authors have addressed the following comments and suggestions.

1. *The authors employed molecular dynamics simulations to generate GeTe amorphous models, more details for generating the amorphous models are therefore necessary to provide.*

The authors are suggested to reference a piece of previous work (J. Mater. Chem. C 11 (2023) 1360-1368) on amorphous phase change materials which provide useful information for generating amorphous models.

Response

We have provided the following additional computational details about the generation of the amorphous models in the Methods section:

“We considered four models with 512 atoms in a cubic simulation box. The atomic densities of the three models were 0.0315, 0.0335, 0.0345 and 0.0365 atoms/Å³, respectively. The amorphous phases were generated by a melt-quench process. The models were first heated to above 2500 K for full randomization. Then the systems were equilibrated at 1000 K for 200 ps, after which they were quenched to 300 K with a quenching rate of 5 K/ps⁵⁶. Finally, the systems were annealed at room temperature for 1 ns.”

We have also added the reference suggested by the reviewer.

Furthermore, we have included new ab initio molecular dynamics simulations for GeSe to address point 9 by referee 4. We have also added computational details for these simulations:

“The MD simulations for GeSe were carried out using the CP2K package⁵⁹. We employed the second-generation Car-Parrinello method⁶⁰ implemented in the QUICKSTEP module of this package. A double-zeta plus polarization Gaussian-type basis set was used to expand the Kohn-Sham orbitals and plane waves with a cutoff of 300 Ry were used to expand the charge density. We employed the same functional used for the generation of the NN potential for GeTe⁵⁶. We considered two models with 512 atoms in a cubic simulation box at atomic density 0.042 and 0.046 atoms/Å³, respectively. The amorphous phases were generated with a similar melt-quench process as for GeTe, albeit with shorter quenching and annealing times. The models were first heated to above 2500 K for full randomization, then they were equilibrated at around 1000 K for 30 ps, after which they were quenched to 300 K with a quenching rate of 15 K/ps and annealed at room temperature for 30 ps. The time step was set to 0.002 ps.”

2. The calculated pair distribution function $g(r)$ of GeTe amorphous in Fig. S4 is different with the $g(r)$ obtained by experiment. The intensity of the second nearest neighbor peak in the simulation is weaker, manifesting that the amorphous GeTe in simulation is more disordered than that in experiments. The authors should carefully check the GeTe amorphous models, which will influence the results of electronic structure. Moreover, it is easy to obtain the $S(Q)$ by molecular dynamics frames. It might be more convincing to compare the $S(Q)$ of experiments and calculations.

Response

We thank the reviewer for the valuable comment. Below (Figure R6 and Figure S10a) we compare the experimental and computational $S(Q)$ at zero pressure. There are some quantitative discrepancies: in particular, the computational first peak is lower than the experimental one and is shifted to higher Q . Nevertheless, such discrepancies are in line with those found in the literature between experimental $S(Q)$ and those obtained from ab initio melt-quenched models of PCMs based on standard DFT functionals. Qualitatively, the

simulations reproduce the main features of the experimental $S(Q)$, including the presence of the pre-peak. In Figure R7, we also show the experimental and theoretical $S(Q)$ at two different pressures. This figure has been included in the supplement as Figure S10.

Figure R6. Experimental and computational total structure factor $S(Q)$ of GeTe without pressure.

Figure R7. Pressure dependence of the total structure factor $S(Q)$ of GeTe obtained from (a) experiment and (b) MD simulation. The simulations qualitatively reproduce the pressure dependence of $S(Q)$ obtained from the experiments, including the vanishing of the pre-peak around 1.0 \AA^{-1} .

3. It seems that the pseudo-bandgap of GeTe amorphous models is caused by the functionals. Using a better functional, e.g., HSE06 or PBE0, the author could obtain more precise electronic structures, which is closer to the situation in experiments.

Response

We have repeated the calculation of the density of states of amorphous GeTe using the hybrid functional *HSE06*. It turns out that the results are in line with the ones shown in the manuscript, which were obtained with the recently developed TASK functional (Ref. 44). This semilocal, kinetic-energy density-dependent functional has been shown to yield accurate energy gaps at a fraction of the computational cost of hybrid functionals.

In the Supplementary Information, we have included a plot containing both the TASK and HSE06 density of states for a direct comparison (Figure S14). The plot is also shown below.

Figure R8. Comparison between the density of states of GeTe at two different pressures calculated using the TASK functional (continuous lines) and the hybrid HSE068 functional (dashed lines). The results hardly depend on the functional. The same basis set and cutoffs were used in the two sets of calculations. The hybrid-functional calculations were accelerated by using the auxiliary density matrix method (ADMM) (Guidon *et al.*, *J. Chem. Theory Comput.* 2010, 6, 2348–2364).

Reviewer #4:

In the manuscript entitled “Pressure-induced reversal of Peierls-like distortions elicits the polyamorphic transition in GeTe and GeSe”, Fujita et al. have studied a polyamorphic transition in GeSe and GeTe through high pressure synchrotron X-ray experiments and molecular dynamic simulations. The key focus of the study is to investigate if modulations in Peierls-like distortions elicit a polyamorphic transition in GeSe and GeTe and if so, how they can be tuned to adjust the transition. Having carefully evaluated the manuscript, I am unfortunately unable to recommend it for publication in Nat. Commun. as there are many concerns I have about the manuscript which are listed below, do not go with the high standard of Nat. Commun.

1. How do the authors justify calling the high-pressure phase a distinct amorphous phase? Especially for GeTe, isn't it possible that beyond the so called Paa, the phase is macroscopically amorphous but with microscopic crystallization. All the characterizations provided, viz. reduced Peierls distortion, higher coherence length, reduced compressibility etc. would be observed even if a material is undergoing a gradual transition from an amorphous to crystalline phase. Also how did the authors estimate the partial crystallization and quantify it? In addition, the authors must study the extent of crystallization at different pressures and see the change in trend near the proposed polyamorphic transition temperatures.

Response

With the high photon flux at the high-energy synchrotron X-ray source, we were able to collect high-resolution diffraction data, allowing us to clearly identify the crystalline and amorphous phases. Figure R9a below shows an example of the diffraction profile of GeTe. The peaks observed around the scattering angles, $2\theta = 2.45$ degree and 3.5 degree, exhibit a ten times narrower width of about 0.04 degree, compared with ~ 0.4 degree of the broad peak located at 2.4 degree. The width of the diffraction peak is a key feature to distinguish the amorphous phase from the crystalline phase in diffraction profiles since the width is inversely related to the length scale of the structural order of materials. Longer the structural order, narrower the width. For example, the sharp Bragg peak at 2.45 degree indicates the existence of crystals with long-range periodic structure longer than 100 nm, while the broad peak at 2.4 degree indicates an amorphous structure with coherent length of ~ 5 Å. Thus, the high- (and low-) pressure state of the amorphous phase can be clearly distinguished from the crystalline state.

In addition, the Rietveld refinement is a well-established method in crystallography to identify the crystalline phase and determine the structure. We added the details of the Rietveld refinement and the representative results in the Supplementary Information. As shown in Fig. 1 and Fig. S2, by using this method, the partially crystallized GeTe data were modelled with the cubic and rhombohedral phase of crystalline GeTe. The broad amorphous scattering below the crystalline Bragg peaks were expressed as the sum of 9 profile functions to fit the data. An excellent agreement was obtained between the observed and calculated diffraction profiles. For example, the difference between the observed and calculated intensity was only 0.2 % on average for the first 16 peaks of GeTe at 10.2 GPa, which constitutes ~ 80 % of the total crystalline contribution, suggesting that the cubic model describes the Bragg peaks with sufficient accuracy. The agreement for the broad amorphous peaks was evaluated by the reliability factor of weighted profile R_{wp} , which is commonly used in the analysis of powder diffraction data to quantify the goodness of fitting for the overall analytical range. The resulting R_{wp} of 4.18 % suggests that the broad amorphous peaks are successfully expressed by the assumed 9 profile functions. These results indicate that we can successfully decompose the diffraction data into the crystalline and amorphous contributions.

Furthermore, the high-flux synchrotron X-rays used in the present study are extremely sensitive to tiny amounts of crystalline particles. The absence of Bragg peaks of crystalline GeTe below 3.4 GPa demonstrates that the partial crystallization occurred at 3.4 GPa. Thus, the sample did not undergo a gradual transition from amorphous to crystalline state below

3.4 GPa. Figure R9b shows the pressure dependence of the normalized intensity sum of crystalline and amorphous phases, obtained from the results of the Rietveld refinement. The normalized sum (blue triangles) is proportional to the volume of the crystalline fraction of the materials. The fluctuation and error bars above 3.4 GPa can be partially attributed to the shift of the X-ray beam positions on the sample after increasing the pressure. The increase of normalized intensity sum is about 20-30% during compression above 3.4 GPa, indicating a rather stable crystalline volume fraction.

Based on the evidence given above, we are confident that the high-flux synchrotron X-rays diffraction data are sufficient to distinguish the crystalline phase from the amorphous state, and the material did not undergo a gradual transition from amorphous to crystalline phase.

We have added a more detailed description of the data analysis in the Supplementary Information.

Figure R9. (a) The representative diffraction data of amorphous GeTe at 10.2 GPa. The observed raw $I(Q)$ (black) and the extracted amorphous contribution (orange). The amorphous contribution was determined by the Rietveld refinement. The Bragg peaks of the crystalline state show substantially sharper width with respect to the broad amorphous peaks. (b) The pressure dependence of the normalized intensity sum of the crystal and amorphous contributions. The intensity sums of the amorphous contributions are normalized to the result of ambient pressure, while that of the crystalline contributions are normalized to 3.4 GPa. The increase of intensity sum of crystalline contribution is about 20-30% during compression above 3.4 GPa, indicating a rather stable crystalline volume fraction.

2. The authors should provide why pressure acts to reduce the Peierls-like distortion in the studied systems which is the centric part of this manuscript. Unfortunately, no such discussions have been in the present work.

Response

In general, Peierls-like distortions lead to larger atomic volumes than symmetric structures. Intuitively, this stems from the fact that the longer bonds are more easily shortened than the shorter bonds are elongated. This effect is responsible e.g. for the negative thermal expansion observed in tellurium-based liquid alloys (Otjacques et al., PRL 103, 245901 (2009)), in which Peierls distortions disappear at high temperature.

We have added a brief discussion regarding this point in the main text (Page 11, the first paragraph).

3. The motivation of the manuscript is very unclear and confusing. Well, in many places in the manuscript, it is reflected that the polyamorphic transition is an outcome of the suppressing Peierls-like distortion. What is the chemical insight of it? Why Peierls-like distortion is responsible for such a phenomenon?

Response

A much-discussed and remarkable anomaly of disordered materials such as water, silicon, germanium, silica and the Phase-Change Materials (PCMs) is that they exhibit two distinct liquid states and possibly two corresponding amorphous solids, associated with drastic changes in their structure and properties. The transition between the two states (liquid-liquid transition and polyamorphic transition) has been intensively studied and debated; however, the relation between these two transitions is unclear (Tanaka, *J. Chem. Phys.*, 2019, 153, 130901). Thus, deeper understanding of this relation can provide insight for a unified description of liquid and amorphous polymorphism.

Particularly for PCMs, a recent experimental and computational study revealed that the atomic-scale mechanism of the liquid-liquid transition (LLT) (Zalden *et al.*, *Science*, 364, 1062-1067) is associated with the onset of Peierls-like distortion in the local structure induced by decreasing temperature. Thus, based on the hypothesized relation between the LLT and polyamorphic transition, we can naturally speculate that the polyamorphic transition between two amorphous solids under pressure can be understood with a similar mechanism to the one established for the LLT. This is a crucial point to be verified. In this work, we have demonstrated clear evidence of the suppression of the Peierls-like distortion underlying the polyamorphic transition. This is effectively a reversal of the structural effect observed in concomitance with the metal-semiconductor transition induced by supercooling in PCMs. This suggests the existence of an as yet unidentified polyamorphic/LLT transition line on the P-T metastable phase diagram (Figure 5c), where the Peierls-like distortion can be tuned by both pressure and temperature to tune the properties of the materials.

The chemical insight of the suppression of Peierls-like distortion can be understood in terms of the delocalization of electrons. A lower degree of Peierls-like distortion corresponds to less covalent bonding characters of amorphous GeSe and GeTe (towards more metallic-like). The distorted local structure with long and short bonds changes towards the ideal octahedral coordination under compression, increasing the coordination number in the 1st coordination shell. Thus, the electrons localized around the three covalent short bonds in the low-pressure state will be more likely shared with the additional short bonds at the high-pressure states, resulting in more delocalized and metallic-like bonding characters. The physical picture is supported by the observation of the closing of pseudo-bandgap under high pressure, revealed by the electronic DOS plot (Figure 5b).

4. Assuming that there is indeed a polyamorphic transition, how are the authors confidently stating that these transitions are 'driven' by the vanishing of Peierls-like distortion? The reduction could be just an effect of pressure on an amorphous phase. For instance, during LLT, if one observes a change in viscosity, one can't attribute it to be the driving force for the LLT.

Response

Thank you for pointing out this wording issue. The choice of "driven" may not have been suitable in the circumstance. In the text, we have rephrased the sentences to clarify that the vanishing of Peierls-like distortion is the underlying microscopic structural changes of the polyamorphic transition.

5. Authors have stated that "With increasing pressure, the sharp Bragg peaks of GeTe appear at 3.4 GPa, indicating partial crystallization, while GeSe remains fully amorphous up to 10.0 GPa." How do the authors evaluate sharpness? I can also see sharpening of the $Q(\text{\AA}^{-1}) \sim 2.4$ peak in Fig. 1a up to some extent at 10 GPa. How did they confirm the amorphous to crystal transition just based on this?

Response

Regarding the distinction between amorphous phase and crystalline states, please see our detailed response to your Question 1. As shown in Figure S5, we have evaluated the sharpness of the amorphous diffraction peak by estimating the full width of half maximum (FWHM). As the reviewer pointed out, the FWHM of the diffraction peak of GeSe at 2.4\AA^{-1} decreases from 0.75\AA^{-1} at ambient pressure to 0.6\AA^{-1} at 10.2 GPa. Considering the diffraction profiles of GeTe, the crystalline Bragg peaks should show an FWHM below 0.1\AA^{-1} , which is at least 5 times narrower than that of the peak at 2.4\AA^{-1} . Furthermore, if GeSe crystallized in the stable orthorhombic phase (space group: Pnma), we would have seen at least 5 Bragg peaks in the Q-range from 1.8\AA^{-1} to 4.4\AA^{-1} (von Rohr et al., *J. Am. Chem. Soc.*, 2017, 139, 2771-2777). However, no peak is observed in this Q-range. Therefore, the sharpening of the peak results from the increased coherent length of the amorphous structure by 0.7\AA under compression, not from crystallization.

6. Since $S(Q)$ and $g(r)$ are related via a Fourier transform, can the authors confirm if their simulations can reproduce the pre-peak feature and its vanishing with pressure?

Response

We have computed the $S(Q)$ for two of our models and an additional model at zero pressure. The figure is shown below. The simulations reproduce the pre-peak at low pressure, whereas at high pressure this feature disappears. This figure has been included in the Supplementary Information as Figure S9.

Figure R10. $S(Q)$ of amorphous GeTe at zero pressure and at pressure 0.59 GPa and 1.97 GPa. The low-pressure curves exhibit a pre-peak, whereas the pre-peak disappears at 1.97 GPa due to the suppression of Peierls-like distortion.

7. In Fig. 2c, what are the physical significances of the fitting parameters α and γ ? It is quite clear from the figure itself that the decay is much faster in GeTe than GeSe, therefore there's no need of the fitting the data unless the fitting parameters provide some valuable relevant information.

Response

The fitting is used to quantify the rate and the shape of decay of the pre-peak intensities. We agree with the reviewer that it is clear that the decay is much faster in GeTe than GeSe from the figure itself in this specific case. However, future studies may want to investigate the temperature effect on the pre-peak intensity under high pressures. For instance, instead of restricting oneself to room temperature, one could increase the temperature up to the glass transition T_g and study the pressure-dependence of the pre-peak intensity at each temperature. The difference in the decay behavior at different temperature might not be so obvious from visual inspection, which necessitates a quantitative comparison. As illustrated in Fig.5c, there is a large temperature-pressure space to explore for better understanding the effect on the local structure. Furthermore, if one studies the pseudo-binary compositions on the GeTe-GeSe tie line, a quantitative analysis of the pre-peak intensity would also be necessary to understand the effect of small compositional changes. Thus, we think that the fitting parameters of an empirical model are useful for quantitative comparison in future related work.

8. What do the peaks correspond to in $I(Q)$ plots shown in Fig.1. In Fig. S4 (a), why is the $g(r)$ intensity decreasing with increasing pressure? Also, in Fig 3. (a) and (b), the number on y-axis is meaningless since it's a stacked plot.

Response

The $I(Q)$ peaks in Fig.1 result from the diffuse scattering of the amorphous structures. Unlike diffraction patterns (e.g. Bragg peaks) seen in crystalline materials, it is not straightforward to assign each diffuse scattering peak in reciprocal space to distinct structural attributes, as amorphous structures lack long-range order and translational symmetry. The modelling techniques, such as the Rietveld refinement, commonly used to model Bragg peaks for crystals, cannot be implemented for amorphous structures. Thus, for amorphous materials, it is helpful to perform total scattering measurements (i.e. including contributions of both Bragg peaks and diffuse scattering with large- Q range) and pair distribution function analysis, to study the structure in real space. One of the important contributions of this work is the experimentally determined total scattering with a wide Q -range (up to 27 \AA^{-1}), which allows to extract high-resolution $G(r)$ in real-space by Fourier transform of $I(Q)$ (see Methods and SI). As shown in earlier studies, the ratio of $R=r_2/r_1$ of $G(r)$ (or $g(r)$) reflects the degree of Peierls-like distortion, since the length of some long bonds of the Peierls-distorted structure can be significantly longer than the first coordination shell, and even reaches the second shell (Zalden *et al.*, *Science* 364, 1062-1067, 2019). In Fig. S7(a), the decreasing peak height of $g(r)$ around the 1st and 2nd peaks is compensated by the rise of the in-between area of two peaks around 3 \AA . Together with the change of peak positions, this implies that long bonds are significantly compressed, and the length distribution of long and short bonds overlap significantly after the suppression of the Peierls-like distortion. The relation between the shape of the $g(r)$ peaks and the evolving distribution of bond length has been widely confirmed in various types of Phase-Change Materials (Zalden *et al.*, *Science* 364, 1062-1067, 2019, or Xu *et al.*, *Adv. Electron. Mater.* 2015, 1, 1500240).

Thanks for pointing out the issue of the y-axis in Fig.3. We have added a note in the figure caption to clarify that the curves have been vertically shifted for clarity.

9. *It is really surprising that the authors haven't provided any molecular dynamic simulation studies for GeSe.*

Response

We did not provide simulations for GeSe because we do not have a neural-network potential for this compound. We understand the referee's concern and we have performed expensive ab initio molecular dynamics simulations for amorphous GeSe to investigate its behavior as a function of pressure. We studied 512 atoms models at atomic density of 0.042 and 0.046 atoms/ \AA^3 corresponding to pressure of 2.42 and 5.17 GPa. We considered only two pressures due to the high computational cost of ab initio simulations. The resulting ALTBC plots are shown in the supplement (Figure S11 and Figure R11 below). They show that, at 2.42 GPa, there is still pronounced Peierls-like distortion and that the distortion has mostly disappeared at 5.17 GPa. Thus, a polyamorphic transition occurs in GeSe as well but at higher pressure with respect to GeTe, in agreement with experimental data. Besides including Figure S11 in the supplement, we have also summarized the main results in the paper and we have discussed the computational details in the methods section.

Figure R11. Angular limited three body correlation (ALTBC) plots of GeSe at 2.42 GPa (left) and 5.17 GPa (right) showing suppression of Peierls-like distortion for increasing pressure. Comparison with the GeTe data (Figure 5 in the main text) indicates that the polyamorphic transition occurs at higher pressure as compared to GeTe, in agreement with experimental data.

10. Since GeSe exhibits strong covalent bonding environment and the bond strength in GeTe is lower compared to GeSe, shouldn't the bulk modulus be lesser in value for GeTe than GeSe?

Response

This is an interesting question. GeSe exhibits higher bonding strength than GeTe at ambient conditions. Indeed, in the low-pressure states, the bulk modulus of amorphous GeSe (18.8 ± 0.7 GPa) is significantly higher than that of amorphous GeTe (8.7 ± 6.8 GPa), as expected.

However, the bonding character can be significantly modified under compression. A previous study revealed that crystalline GeSe shows much less covalent (more metallic) bonding character at elevated pressure, resulting from the suppression of the Peierls-distortion and the associated delocalization of electrons (Xu *et al.*, J. Phys. Chem. C, 121, 25447-25454, 2017). In the present study of amorphous states, we observe the suppression of Peierls-like distortion and the closing of the pseudo-bandgap as footprints of the increase in electron delocalization. We can thus expect that both GeSe and GeTe show significantly less covalent (more metallic) bonding in the high-pressure states. GeSe may even become less covalent than GeTe at high pressure. This may explain the higher bulk modulus of amorphous GeTe (52.9 ± 6.9 GPa) as compared to amorphous GeSe (45.9 ± 2.3 GPa) at high pressures. To this end, we believe that it is not possible to draw conclusions about the high-pressure phases based on the bonding properties at ambient pressure. The study of bonding in the amorphous phases at high-pressure would be an interesting topic for future theoretical work.

REVIEWERS' COMMENTS

Reviewer #1 (Remarks to the Author):

This is a very solid work that thoroughly investigated and understood the glass-glass transition in PCM, and all my questions have been addressed in the revised manuscript. Hence, I support the publication as it is.

Reviewer #2 (Remarks to the Author):

Thanks to authors for their detailed responses and revised version. I recommend the revised version for publication without any further minor/major issues from my side.

Reviewer #3 (Remarks to the Author):

The authors have carefully revised the manuscript and I believe it is ready to publish.

Reviewer #4 (Remarks to the Author):

After carefully considering the revised manuscript and the responses provided by the authors, I am glad to recommend the article for publication. I am pleased with the responses provided by the authors and the revised manuscript appears to stand on a strong scientific footing backed with rigorous analysis. The work is novel in attempting to draw a connection between polyamorphic transition and degree of Peierls distortion in a PCM. However, I recommend the authors further highlight their work's novelty considering a similar recent publication (Adv. Funct. Mater. 2023, 2304926a).

Response Letter

We thank all four Referees for their review reports. All four reviewers are satisfied with the revised manuscript and recommended the article for publication; Reviewer #4 raised a minor point. We provide our point-by-point response to the comment of Reviewer #4 below.

Response to Reviewer #4:

After carefully considering the revised manuscript and the responses provided by the authors, I am glad to recommend the article for publication. I am pleased with the responses provided by the authors and the revised manuscript appears to stand on a strong scientific footing backed with rigorous analysis. The work is novel in attempting to draw a connection between polyamorphic transition and degree of Peierls distortion in a PCM. However, I recommend the authors further highlight their work's novelty considering a similar recent publication (Adv. Funct. Mater. 2023, 2304926a).

Response

We have added the recommended publication by Reviewer #4 to the reference. We also added a sentence below to the main text to briefly discuss the reported study in relation to our results.

“This is in a qualitative agreement with a very recent computational paper, in which the suppression of the distortions is reported to occur at even higher pressure, possibly due to the absence of van der Waals corrections in the simulations.”